# Parametric Assessment of Soil Nailing on the Stability of Slopes Using Numerical Approach

**Tausif E Elahi [1], Md Azijul Islam [1,2,\*] and Mohammad Shariful Islam [1]**

[1]  Department of Civil Engineering, Bangladesh University of Engineering and Technology (BUET), Dhaka 1000, Bangladesh; tausif@ce.buet.ac.bd (T.E.E.); msharifulislam@ce.buet.ac.bd (M.S.I.)

[2]  Department of Civil Engineering, The University of Texas at Arlington, 701 W Nedderman Dr, Arlington, TX 76019, USA

\*  Correspondence: mdazijul.islam@uta.edu or azijul@ce.buet.ac.bd; Tel.: +1-(817)-821-2598

**Abstract:** This study focuses on the stability analysis of slopes reinforced by soil nailing. The effects of slope geometry and nail parameters on slope stability are investigated using PLAXIS 2D. Four different slope angles and three different backslope angles are considered for assessing the effect of slope geometry on the stability of a nailed slope. The factor of safety (FS) was found to decrease with the increasing values of the slope angle as well as the backslope angle. The influence of different nail parameters (nail inclination, nail length, and nail spacing) was also investigated. With the increase in nail inclination, FS was found to increase initially and thereafter, reaching a peak value followed by a drop in FS. The optimum nail inclination was found between 0 and 25° at a horizontal angle, depending on the different slope geometries, which is evident from observation of the slip surface as well. With the increase of nail length, FS increases; however, the increase was small after L/H (length of nail/height of slope) reached a value of 0.9. Moreover, increasing the length of the nail was found to be effective in reducing the lateral movement of the slope. The maximum nail forces are observed in the bottom-most row of nails and increase with the depth. The inclusion of soil nailing with optimum nail parameters can increase FS by 29–75% depending on the slope geometry, signifying the effectiveness of nailing.

**Keywords:** slope stability; soil nailing; factor of safety; nail inclination; optimum nail layout





## 1. Introduction

Soil nailing is widely used as a form of reinforcement to improve the stability of steep slopes and vertical cuts [1–6]. The stability of nailed slopes depends on the mechanism of transferring resisting tensile forces generated in the nails into the ground through friction or adhesion mobilized at the interfaces. Ground movement is restrained by the friction between the nail and the soil. The stability of the nailed slope is governed by various factors such as slope geometry, nail parameters, the soil–nail interaction, etc. [7]. Slope angle, backslope gradient, nail inclination, nail length, the spacing of the nails—these are some of the important parameters that directly affect the stability of slopes. The resistance provided by soil nails is subjected to the orientation of nails with respect to the failure surface, which is dependent on slope characteristics. Moreover, the soil–nail interaction is another important factor for providing shear resistance to stabilize soil mass. However, for engineering purposes, the overall stability of the nailed slope is a prime concern rather than the mechanical behavior of a single nail.

Studies have been carried out by researchers to establish the soil-nail behavior and stability of slopes [8–10]. The limit equilibrium method is adopted by various researchers for assessing the performance of a nailed slope [11–14]. The limit equilibrium model cannot simulate the interaction between the nail and surrounding ground, which affects the mobilization of the resistance provided by nails. Such limitations can be overcome

using numerical methods such as finite element (FE) analysis, which can model the interaction between various elements and predict actual behavior accurately to serve as a design and analysis aid for nailing [2,15,16]. Numerical simulations using FE analysis can provide accurate results if the parameters representing soil characteristics are properly evaluated and interaction among different elements such as nails and soils are correctly considered [17,18]. Unterreiner et al. [19], Babu et al. [20], and Cheuk et al. [21] applied 2D FE or finite difference methods for the analysis of nailed slopes, and many researchers have used this method for different applications, including the undrained capacity and stability of footing lying on layered slopes as well [22–26]. Moreover, various researchers have used this method correctly to predict the soil deformation, stress, nail forces, and factor of safety of the slopes [27–30].

From an engineering point of view, the parameters that have a greater influence on the stability of a soil-nailed slope should be properly investigated for optimum design. The traditional design method of the nailed slope was based on the slope angle, uniform nail length, equal nail spacings, and nail inclination between 10° and 20° [31]. Jaiswal et al. [3], based on their study, stated that the length and inclination of the soil nail are the major controlling parameters for stabilizing the slope. However, the stability of a soil-nailed slope is closely related to other parameters such as the back slope inclination ($\alpha$), cohesion, and friction angle of the soil [1,32].

The performance of the nailed slope has been analyzed both numerically and experimentally to study the different nail parameters and soil–nail interactions under various loading conditions [4–6,33–36]. Tang and Jiang [30] reported the highest FS when the inclination of the nail was 15° with the horizontal in LEM. On the other hand, Sabhahit et al. [37] conducted a numerical study and stated that the horizontal nail is the optimal direction except for the lower-most nails. However, Rotte et al. [7] showed that with the horizontal backslope, the optimal nail inclination increases with the decrease in slope inclination; however, it increases with an increase in backslope inclination. Tang and Jiang [38] suggested that with the increase of nail spacing, FS gradually decreases. Nevertheless, Fan and Luo [1], through FE analysis, stated that the effect of the arrangement of vertical nail spacing is insignificant when the number of nails used in the slope is unchanged. Gunawan et al. [39] correlated the length and diameter ratio of the nail with FS for 20 m height of slope for different internal friction angles and slope angles in LEM. They concluded that with the increase of the diameter and the length of nail, FS increases. On the contrary, Fan and Luo [1] suggested that the length of the nail, which is located at the bottom one-third of the slope, is more important in the stability of slopes than that situated in the upper portion. Tei et al. [40] performed centrifuge model tests, and all slopes were collapsed due to the insufficient anchorage length of the nails beyond the failure surface. Specifically, external failures occurred for short lengths and dense spacing, while internal failures occurred for long lengths and sparse spacing [41]. Pradhan et al. [42] analyzed the pullout resistance of the grouted nail and stated that it increases linearly with an increase in vertical overburden pressure. Moreover, shear strength parameters such as cohesion (c) and internal friction angle ($\varphi$) of soil have great influence on the stability of the nailed slope. Wu et al. [43] investigated the influence of c and $\varphi$ values on a nailed slope though the FE model and showed that the factor of safety of slope reinforced with nails largely depends on c and $\varphi$. In addition, the selection of a proper soil model for FE analysis is essential. Singh and Babu [44] studied the behavior of nailing using different types of soil models such as the hardening soil model and Mohr–Coulomb (MC) soil model in Plaxis 2D, an FEM software. They concluded that the MC model can predict the behavior of the soil-nailed slope accurately, as found in other literature [1,45], and recommended the use of advanced models for the analysis of soil–nailed walls constructed in soft soils. Furthermore, Cheng et al. [46] reported nailing has a lesser effect on stabilizing soft clays as it is hard to establish good bond strength, indicating the importance of soil selection for implementing nailing through FE analysis.

The literature review reveals that the stability of the slopes is heavily influenced by variations in different nail parameters' length, inclination, spacing, and types of soil. Previous studies approximated the crest of the slope as horizontal, which is rare in actual practice [44,47]. Very few studies considered the influence of back slope inclination [1,7]. Most of the studies carried out are limited within a few specific slope angles and nail parameters. Therefore, limited knowledge is available considering the combined influence of all the parameters such as slope angle, back slope inclination, nail inclination, nail layout, nail length pattern, and properties of soil. During the devastating landslide of 2017 in Rangamati, Bandarban, and Chattogram—three hilly districts of Bangladesh—a significant loss of human life and significant damage in terms of economy and infrastructure occurred [48–50]. Islam [48] and Islam et al. [50] reported that most of the existing hill slopes in these three districts are vulnerable and vegetation is not effective to increase factor of safety. Hence, the motivation behind the present research work is to study the alternative economical method of stabilizing slopes, and, in this context, the potential of nailing has been assessed to stabilize slopes of varying angles and backslope angles and determine the optimum nail parameters. This study attempts to propose an optimum nail layout for the protection of existing hill slopes in Rangamati, which has been a hotspot for major landslides over the years, and such a study is missing in the existing literature in the context of Bangladesh.

The objective of this study is to evaluate the effect of different soil-nailing parameters: nail inclination, length, spacing, and nail forces on slope stabilization, and to find out an optimum nail layout. Based on existing slope geometry, an FE model is developed using commercially available finite element software, PLAXIS 2D. Finally, a comparative analysis has been carried out between the numerical results obtained from the present study and the test results available in the existing literature, to verify the FE model. The findings of the present study could be a useful guideline for implementing soil nailing as a method of preventing catastrophic slope failure in the hilly regions of Bangladesh.

## 2. Materials and Methods

### 2.1. Soil

For this study, soil samples have been collected from existing hill slopes located in -the Rangamati district of Bangladesh. A typical slope is shown in Figure 1a, where the height of the slope is approximately 10 m and the slope angle is approximately 60°. A landslide failure is shown in Figure 1b, which occurred in June 2017. From the failed slope, it can be observed that the failure starts from the crest of the slope, and then the soil mass moves downward and spreads around the toe of the slope. Typically, the slide height was observed around 7 m to 25 m, whereas the spread of the fall was found in between 5 m to 15 m.

The thin-walled Shelby tube sampler was used in accordance with ASTM D1587-08 for the collection of the undisturbed soil samples (Figure 1c). The grain size distribution [51], specific gravity test [52], Atterberg limit test [53], and direct shear test [54] have been performed according to ASTM standards for the characterization of soil, and the test results are presented in Table 1 along with the grain size distribution in Figure 2. The values of the modulus of elasticity, E, presented in Table 1 are based on the existing literature [55]. Soil is classified as clayey sand (SC) according to the unified soil classification system. The soil properties obtained from the test results have been used for FE analysis.

**Table 1.** Properties of the soil used in this study.

| Property | Parameter | Soil A |
|---|---|---|
| Specific gravity | Specific gravity, $G_S$ | 2.69 |
| Grain size distribution | Effective particle size, $d_{10}$ (mm) | 0.004 |
|  | Average particle size, $d_{50}$ (mm) | 0.075 |

**Table 1.** *Cont.*

| Property | Parameter | Soil A |
|---|---|---|
| Atterberg limits | Liquid limit, LL (%) | 35 |
| | Plastic limit, PL (%) | 25 |
| | Plasticity index, PI (%) | 10 |
| Shear strength parameters | Cohesion, c (kPa) | 10 |
| | Angle of internal friction, $\varphi$ (°) | 35 |
| | Modulus of elasticity, E (MPa) | 25 |
| Soil classification | Unified soil classification system (USCS) | SC |
| Unit weight | Dry unit weight, $\gamma_{dry}$ (kN/m$^3$) | 16.7 |
| | Saturated unit weight, $\gamma_{sat}$ (kN/m$^3$) | 19.0 |

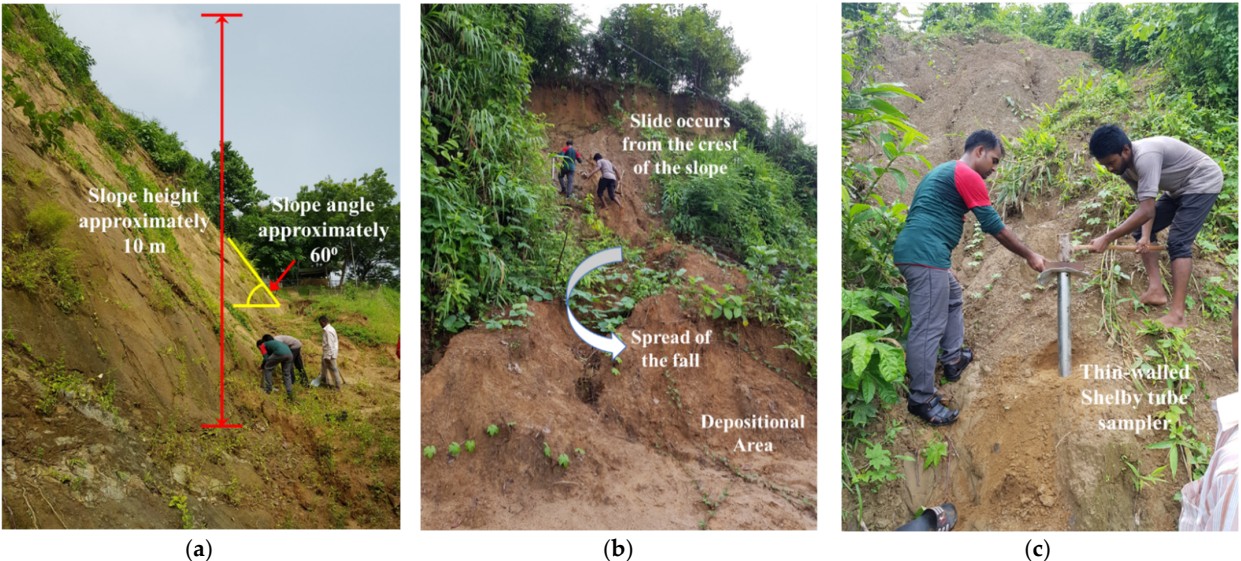

(**a**) (**b**) (**c**)

**Figure 1.** (**a**) Existing hill slope in Rangamati with approximate dimensions; (**b**) a slope failure in Rangamati occurred in 2017; (**c**) soil sample collection using thin walled Shelby tube (all photos were taken by Md Azijul Islam).

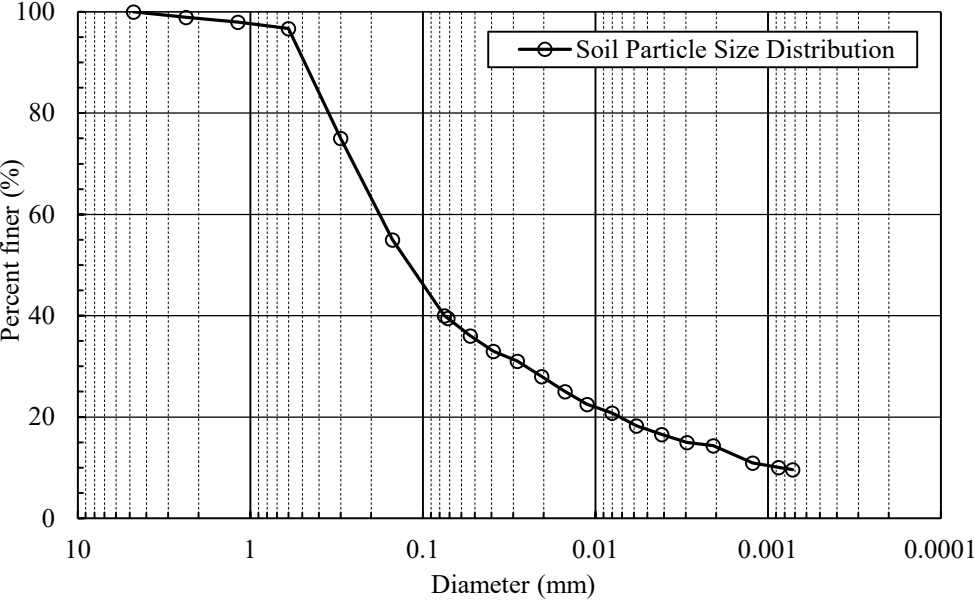

**Figure 2.** Grain size distribution of the soil used in the study.

## 2.2. Finite Element (FE) Model

For engineering practice, the overall stability of the nailed slope is the main priority rather than the behavior of an individual nail. In the present study, PLAXIS 2D, a non-linear finite element program, has been used for calculating the FS of slopes as it takes the mechanical behavior of the soil–nail interaction into account.

A schematic diagram of the geometric model used for FE analysis is shown in Figure 3. A slope height (H) of 10 m is considered in this study as the slope height of the existing site was found to vary within 10–15 m, as reported by Elahi et al. [56]. The nail used in this study is a 29 mm diameter rebar, which is surrounded by a grout diameter of 10 cm. The nail and grout properties were considered based on the availability in the local market. Four slope angles, $\beta$ = 45°, 60°, 75°, and 90°, and three backslope angles, $\alpha$ = 0°, 10°, and 20°, are considered for assessing the influence of slope geometry on the stability of slopes. The slope angles were considered based on the actual field slope angle, which was found to vary within 30°–75°. Google Earth Pro software was used to determine the slope angle, slope heights, and other dimensions. The obtained data from Google Earth were also verified during the field study through field surveying. Nail inclination, $\theta$, varied from 0° to 35° at an interval of 5°, whereas nail length, L, varied within 5 to 12 m for analyzing the effect of these parameters on stability. Moreover, for assessing the effect of vertical nail spacing ($S_v$), four spacings of 1.25 m, 1.50 m, 1.75 m, and 2.0 m are considered, and horizontal spacing, ($S_h$), is maintained 1 m.

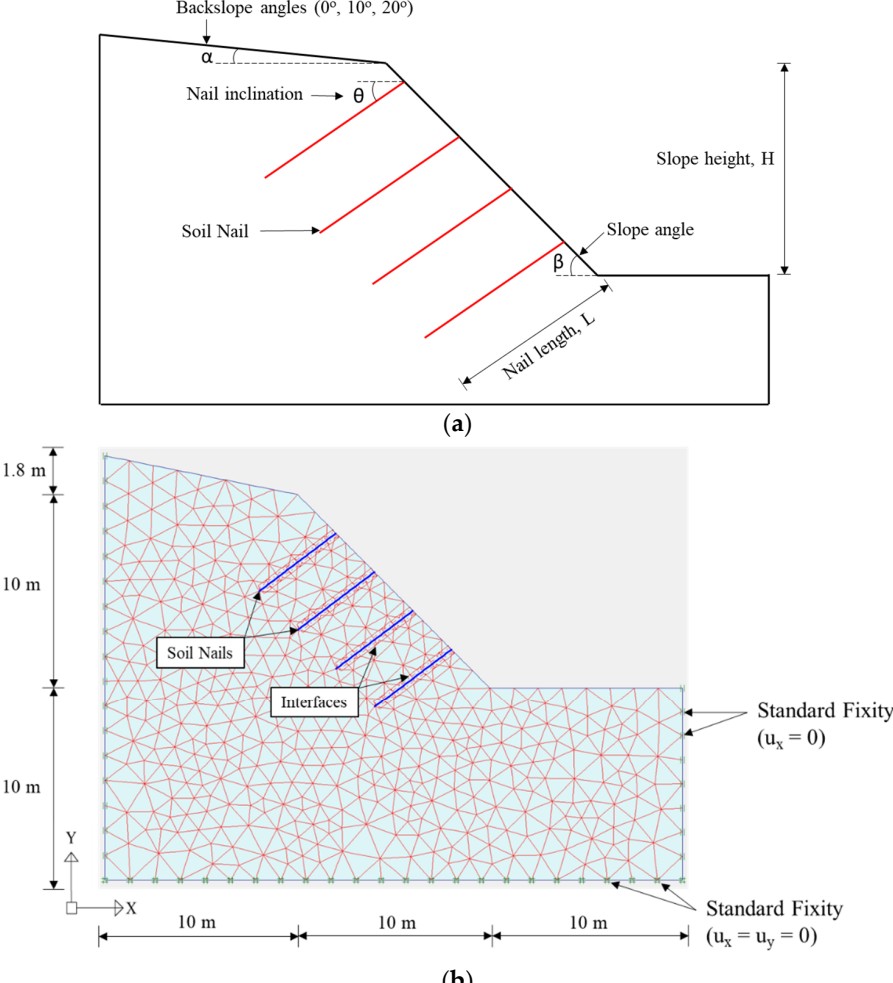

**Figure 3.** (**a**) Schematic diagram of the model; (**b**) PLAXIS model with boundary conditions, interfaces, and fine mesh.

The plane strain model is adopted here for analyzing the slope stability with PLAXIS. In this case, strain in the direction perpendicular to the plane section (which is also the largest dimension) is zero [57,58], and in this stability analysis it extends in a longitudinal direction for a considerable distance, so that a plane strain model is applicable. Fifteen-noded triangular elements are used for modeling, by which failure load and safety factors are correctly predicted, and for the Phi-c type of analysis in PLAXIS, the fifteen-node elements are technically better for the factor of safety calculations. Fourth-order interpolation for displacement and numerical integration involves twelve gauss points in the fifteen-noded triangular element [27], and Rawat and Gupta [45] stated that in the case of analysis that involved nails, anchors, and geogrids, fifteen-noded triangles provide more accurate results compared to those of six-noded elements. The base of the modeled slope is fixed in the x-y direction using the standard fixities, while the back of the slope is restricted only in the x-direction. Moreover, the slope face is allowed to move both horizontally and vertically, and the top of the slope is free to move in a vertical direction. The side faces of the model are allowed to move vertically, restricting the horizontal movement. The Mohr–Coulomb (MC) model was used to model the soil. The linear elastic perfectly plastic Mohr–Coulomb model involves five parameters, i.e., the modulus of elasticity (E) and Poisson's ratio ($\nu$) for soil elasticity, friction angle ($\varphi$), and cohesion (c) for soil plasticity and the angle of dilatancy, $\psi$. The drained soil condition has been used to simulate the FE analysis, and the phreatic level has been placed at the bottom.

In PLAXIS, there are various options for simulating reinforcements such as plate element, geogrids, node-to-node anchors, and fixed-end anchors. Several studies have been performed considering soil nails as a plate element, and the results have been validated with field data [1,37,44]. In this study, soil nails are simulated using an elastic equivalent plate element. The literature reveals that bending stiffness and axial stiffness are important parameters to be considered for the simulation of the nails [1,44,45]. Moreover, while designing nails as a plate element of a circular cross-section, an equivalent flexural rigidity, and axial stiffness, needs to be considered. For obtaining the equivalent modulus of the elasticity of nails, $E_{eq}$, the following equation can be used given by Singh and Babu [44]:

$$E_{eq} = E_n\left(\frac{A_n}{A}\right) + E_g\left(\frac{A_g}{A}\right) \tag{1}$$

where $E_g$ = the modulus of the elasticity of the grout material; $E_n$ = the modulus of the elasticity of the nail; $A$ = the total cross-sectional area of the grouted nail; $A_n$ = the area of the soil nail; and $A_g = A - A_n$ = the cross-sectional area of the grout cover. Knowing the value of $E_{eq}$, the axial and bending stiffness can be calculated using Equations (2) and (3):

$$EA = \frac{E_{eq}}{S_h}\left(\frac{\pi D^2}{4}\right) \tag{2}$$

$$EI = \frac{E_{eq}}{S_h}\left(\frac{\pi D^4}{4}\right) \tag{3}$$

where $EA$ = the axial stiffness, $EI$ = the bending stiffness, $D$ = the diameter of the grout, and $S_h$ = the horizontal spacing of the nail. Moreover, the equivalent plate diameter of the nail, $d_{eq}$, is calculated using the following equation in PLAXIS:

$$d_{eq} = \sqrt{12\frac{EI}{EA}} \tag{4}$$

The properties of the nails are calculated using the above equation and presented in Table 2. These properties have been used in FE analysis. Soil nails are discrete elements, and in the finite element model, a discrete element is substituted by a plate, which is extended to the unit width. The plates in the 2D analysis are simulated as beam elements with the interface at the upper and lower side, and the beam is continuous in the out-plane

direction. A virtual thickness was assigned to the interface element in order to define the material characteristics. A virtual thickness factor ($\delta_{inter}$) of 0.1 is used in the analysis to ensure proper soil–nail interactions, as mentioned in the literature by Rawat and Gupta [45], Jaiswal and Chauhan [57,58] and Islam et al. [59]. While producing the mesh, this factor is multiplied by the element thickness, and the material properties of the interface are the same as the surrounding material. To simulate the pullout resistance of the soil nails, a strength reduction coefficient $R_{inter}$ is used to determine the strength of the soil-nail interface, which can be defined as

$$R_{inter} = \frac{c_{inter}}{c_{soil}} = \frac{tan \ \varphi_{inter}}{tan \ \varphi_{soil}} \tag{5}$$

where $c_{soil}$ and $\varphi_{soil}$ are the cohesion and friction angle, whereas $c_{inter}$ and $\varphi_{inter}$ are the cohesion and the angle of internal friction of the soil-nail interface. Chu and Yin [60] performed several pullout and interface shear tests on cement-grouted nail and surrounding soil to determine the value of $R_{inter}$, and the results indicated that this value exists within 0.95–1.07. As such, $R_{inter}$ = 1 is considered in this study, and a similar value is used by Fan and Luo [1] and Rawat and Gupta [45] in an FEM analysis for the slopes stabilized with soil nails. After the modeling of the soil nails, a mesh is generated, and, in this study, a finer mesh is considered for obtaining accurate results.

**Table 2.** Soil and nail parameters used for FE analysis.

| Parameter | Value |
|---|---|
| Nail element | Elastic Plate |
| Axial stiffness, $EA$ (kN/m) | $2.90 \times 10^5$ |
| Flexural rigidity, $EI$ (kN m$^2$/m) | 181.4 |
| Poisson's ratio, $\nu$ | 0.3 |
| Dilatancy angle, $\psi$ (°) | 0 |
| Interface thickness, $\delta_{inter}$ | 0.1 |
| Interface strength, $R_{inter}$ | 1.0 |
| Modulus of elasticity of nail (GPa) * | 200 |
| Modulus of elasticity of grout (GPa) * | 22 |

* Singh and Babu [44].

*2.3. Calculation of FS*

The FS of slopes in PLAXIS is computed using the phi-c reduction method. In this method, the strength parameters are being reduced until the slope fails. This method provides a value of the incremental multiplier, ΣMsf; as a result, it converges when the slope fails. The ratio of the actual parameter to the critical parameter is considered as FS. At that stage, FS is evaluated based on the following equation:

$$\text{FS} = \Sigma\text{Msf} = \frac{tan\phi_{input}}{tan\phi_{output}} = \frac{C_{input}}{C_{output}} \tag{6}$$

As mentioned earlier, the stability of the slopes is assessed in terms of the FS in this study. Following the procedure discussed so far, FS is calculated for various slope conditions under different nail parameters.

## 3. Results and Discussion

The stability analysis of slopes has been performed for assessing the influence of slope geometry and nail parameters. For studying the influence of slope geometry, different slope angles and backslope angles are considered for the FE analysis. The effect of nail parameters—nail inclination, nail length, and the spacing of the nails and nail force on the stability of slopes—are presented in this section. Moreover, the obtained results from the

analyses are compared with the existing literature for the validation of the results, which are discussed in this section.

### 3.1. Stability Analysis of Unreinforced Slopes

For understanding the effect of nailing on the overall stability of the slopes, a stability analysis of unreinforced slopes was performed. As discussed earlier, stability has been analyzed in this study in terms of factors of safety; therefore, factors of safety for different slope and backslope angles are presented in Figure 4. With the increase in the backslope angle, FS decreases irrespective of slope angles. For horizontal backslope, the maximum FS obtained from the numerical analysis is 2.05 in the case of β = 45°. For practical purposes, it is not always possible to maintain the natural slope in which it is stable, which highlights the need to improve the stability of the existing slopes.

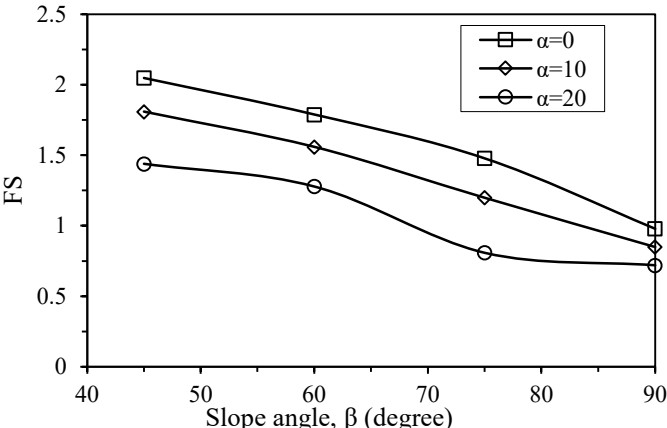

**Figure 4.** Relationship between factor of safety (FS) and slope angle for different backslopes.

### 3.2. Effect of Nail Inclination

The inclination of nails compared to the perpendicular of the potential slip surface at different positions of the slope are different due to the curve failure surface of the slope, thereby indicating the necessity to find optimum nail inclination [1,10]. Numerical analyses are conducted to understand the effect of nail inclination on the overall stability of the nailed slopes for different slope orientations, which are presented in Figure 5.

With the increase of nail inclination, FS increases initially up to a certain inclination for which FS is maximum. The increase of inclination beyond that causes a drop in FS for all the slope and backslope angles. However, for a perpendicular slope (β = 90°), maximum FS is obtained for nail inclination of 0°, i.e., for the horizontal nail, which is the optimum nail inclination. The optimum nail inclination is defined as the nail inclination for which maximum FS is obtained. In the case of horizontal backslope, the optimum nail inclinations for β = 45°, 60°, 75°, and 90° are 25°, 20°, 10°, and 0°, respectively. The optimum nail inclinations for β = 45°, 60°, 75°, and 90° for α = 10° are 30°, 25°, 15°, and 0°, respectively; and for α = 20°, they are 30°, 30°, 20°, and 0°, respectively. Moreover, Figure 5 reveals that for a particular slope angle, an increase of backslope is found to increase the optimum nail orientation, indicating that the optimum nail orientation is highly influenced by the backslope of nailed slopes.

To demonstrate the fact of optimum nail inclination selection, a numerical model of stability analysis is presented in Figure 6. Three cases are shown in Figure 6—a horizontal nail, and nail with an inclination of 20° and 30° for soil with β = 60°. From Figure 5, the optimum nail inclination was found to be 20°. Figure 6a reveals that for the horizontal nail, the nails fail to reach the slip surface, whereas for the 30° nail inclination, nails just reach the slip surface (Figure 6c). On the other hand, with the same length of the nail, with a 20° nail inclination, nails go well beyond the slip surface, which is the requirement for generating nail resistance (Figure 6b), justifying the importance of optimum nail orientation [7].

However, most of the past studies carried out by the researchers stated that the horizontal nail improves the stability of the slopes significantly [31,55]. The results obtained from numerical analyses in this study show the importance of the selection of optimum nail orientation, and findings in this study are in good agreement with the results reported by Fan and Luo [1] and Rotte et al. [7].

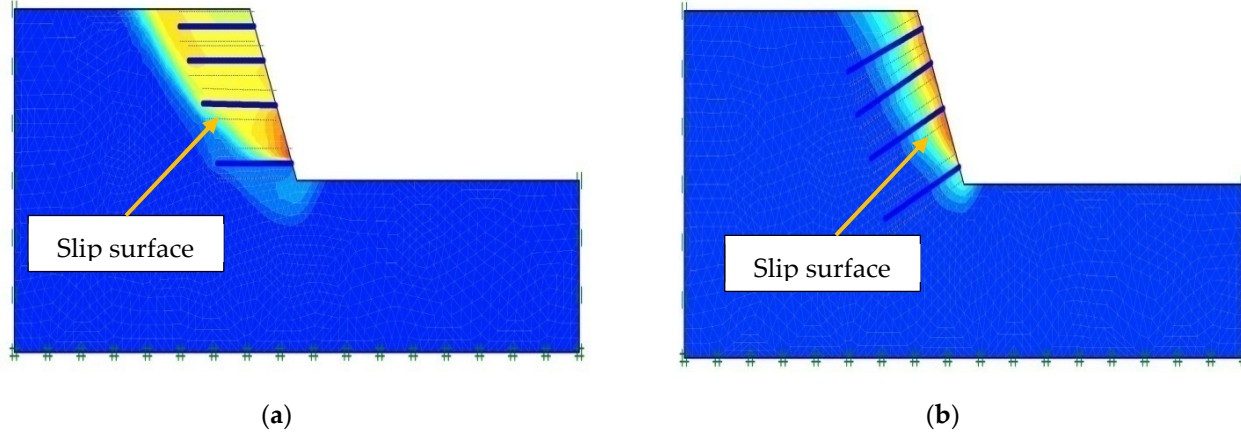

**Figure 5.** Effect of nail inclination on FS of slopes with different backslope inclinations: (**a**) α = 0°, (**b**) α = 10°, and (**c**) α = 20°.

**(a)**            **(b)**

**Figure 6.** *Cont.*

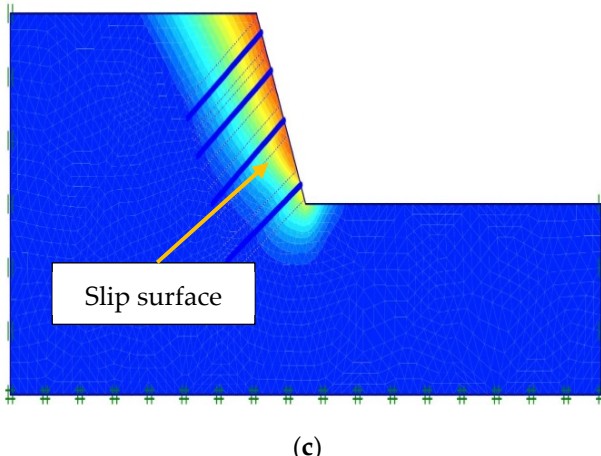

(**c**)

**Figure 6.** Numerical model of analysis for soil with various nail inclinations: (**a**) horizontal nail, θ = 0° (**b**) θ = 20°, and (**c**) θ = 30°.

*3.3. Influence of Nail Length*

The nail length is another important parameter that has a significant effect on the overall stability of the slopes. To study the effect of nail length, various nail lengths have been considered, which have been expressed in terms of L/H. It is to be noted that for the analyses in this section, nail lengths have been varied, keeping the nail inclination fixed at an optimum nail inclination obtained from the previous section. In the case of horizontal backslope, the optimum nail inclinations for β = 45°, 60°, 75°, and 90° are 25°, 20°, 10°, and 0°, respectively. The optimum nail inclinations for β = 45°, 60°, 75°, and 90° for α = 10° are 0°, 25°, 15°, and 0°, respectively; for α = 20°, they are 30°, 30°, 20°, and 0°, respectively. The resistance obtained from the nail is generated due to the available length of the nail exceeding the slip surface [7]. Between the nail boundary and the soil, frictional resistance is produced, which in turn restricts the movement of soil mass, providing the stability of slopes. Figure 7 presents the effect of nail length on FS for different slope arrangements. The length of the nail to slope height ratio, L/H, is varied within 0.50–1.20 for conducting the numerical analysis. With the increase in the value of L/H, FS is found to increase for all the slope angles, and the percent increase in FS varies within 23–63% depending on the L/H and slope angle. Figure 7 depicts that although FS increases gradually for the L/H values up to 0.8–0.9, beyond that value, the increase in FS is small. Therefore, considering the economy and effectiveness of nailing, a nail length of 0.9H can be chosen for stabilization.

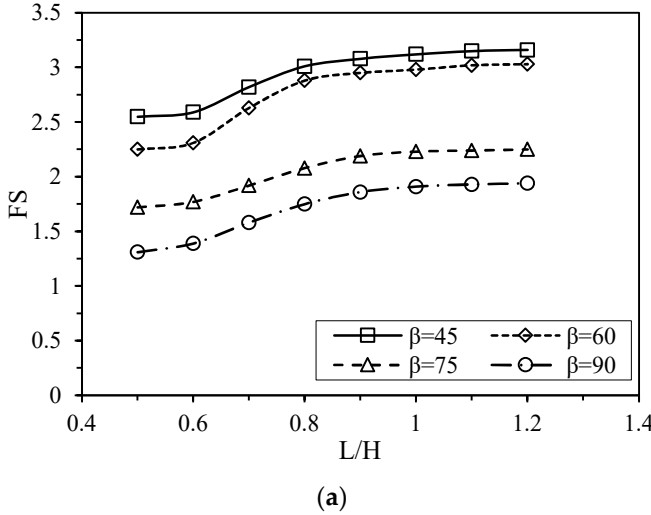

(**a**)

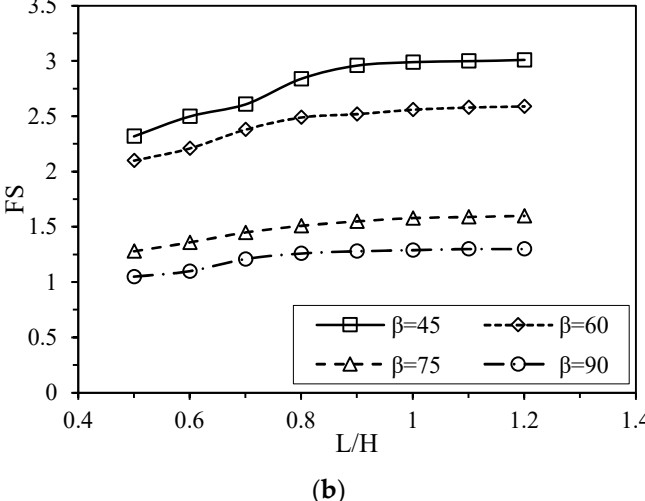

(**b**)

**Figure 7.** *Cont.*

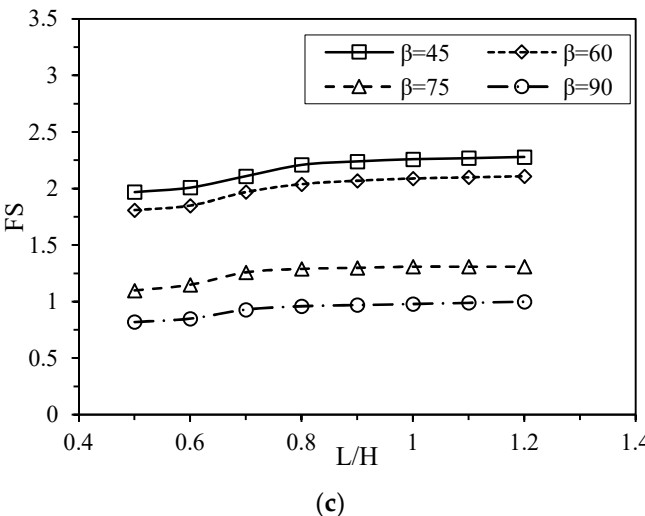

(**c**)

**Figure 7.** Effect of nail length on stability of slopes with different backslope inclinations: (**a**) $\alpha = 0°$, (**b**) $\alpha = 10°$, and (**c**) $\alpha = 20°$.

To justify the selection of 0.9H as optimum nail length, the percent increase in FS when L/H is increased from 0.5–0.9 and 0.9–1.2 is presented in Table 3 for various slope angles with horizontal backslope. The increase of L/H from 0.5–0.9 increases FS by 16.2–33.8% for slopes varying within 45°–90°, whereas the increase of L/H from 0.9–1.2 increases FS by only 1.7–4.3%, which justifies the selection of nail length of 0.9H. Moreover, the relationship between slope top displacement with L/H is also analyzed, which is presented in Figure 8. It is to be noted that the results are obtained from the plastic analysis of the model. It is observed that with the increase of L/H, the slope crest displacement is found to decrease for all the slopes. Similar to the findings in Figure 7, the crest displacement is found to decrease significantly up to L/H = 0.9, and beyond that the increase is small for all slope angles, which indicates that L/H has a distinct relationship with the slope crest displacement. This can be useful while designing soil-nailed slope system.

**Table 3.** Percent increase in FS with values of L/H for horizontal backslope.

| Slope Angle, β (°) | % Increase in FS for L/H (0.6~0.9) | % Increase in FS for L/H (0.9~1.2) |
|---|---|---|
| 45 | 18.9 | 2.6 |
| 60 | 23.7 | 2.7 |
| 75 | 27.7 | 2.7 |
| 90 | 33.8 | 4.3 |

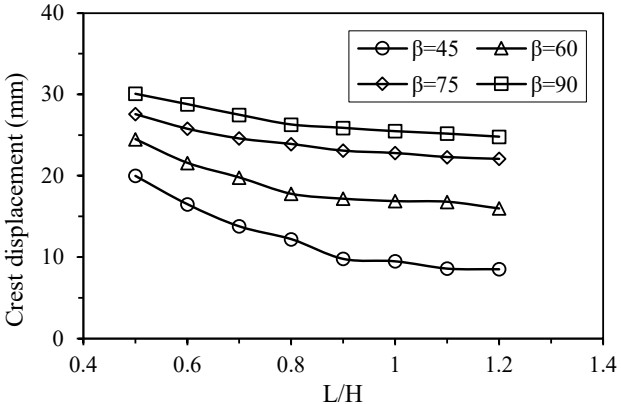

**Figure 8.** Influence of L/H on slope crest displacement with horizontal backslope.

### 3.4. Effect of Vertical Nail Spacing

To investigate the effect of vertical spacing, four nail spacings are considered for numerical analysis: 1.25 m, 1.50 m, 1.75 m, and 2.0 m. It is to be noted that the horizontal spacing of the nail was selected as per the optimum combination obtained from earlier analysis, which was 0.9H. Figure 9 presents the effect of vertical nail spacing on FS for various slope and backslope angles. With the decrease in vertical spacing of the nail, FS is found to increase, and the maximum FS is found for a vertical spacing of 1.25 m. Although reduced vertical spacing improves the FS of slopes, the difference of FS for the spacing of 1.25 m and 2.0 m varies within 9–17%, which is consistent with the findings obtained from Tang and Jiang [37]. Fan and Luo [1] also found a negligible increase of FS for reducing the vertical spacing when the number of nails used in the slope is unchanged and only spacing is varied. Although the reduction of vertical spacing increases FS, considering the economy and construction procedures, it is not always feasible to provide the lowest vertical nail spacing. Hence, the difference in FS for all the analyzed vertical spacing is small; any vertical spacing within 1.25–2.0 m can be chosen for providing nails. Here, in this study, for the further analysis, the vertical spacing of 1.5 m was considered for soil nails.

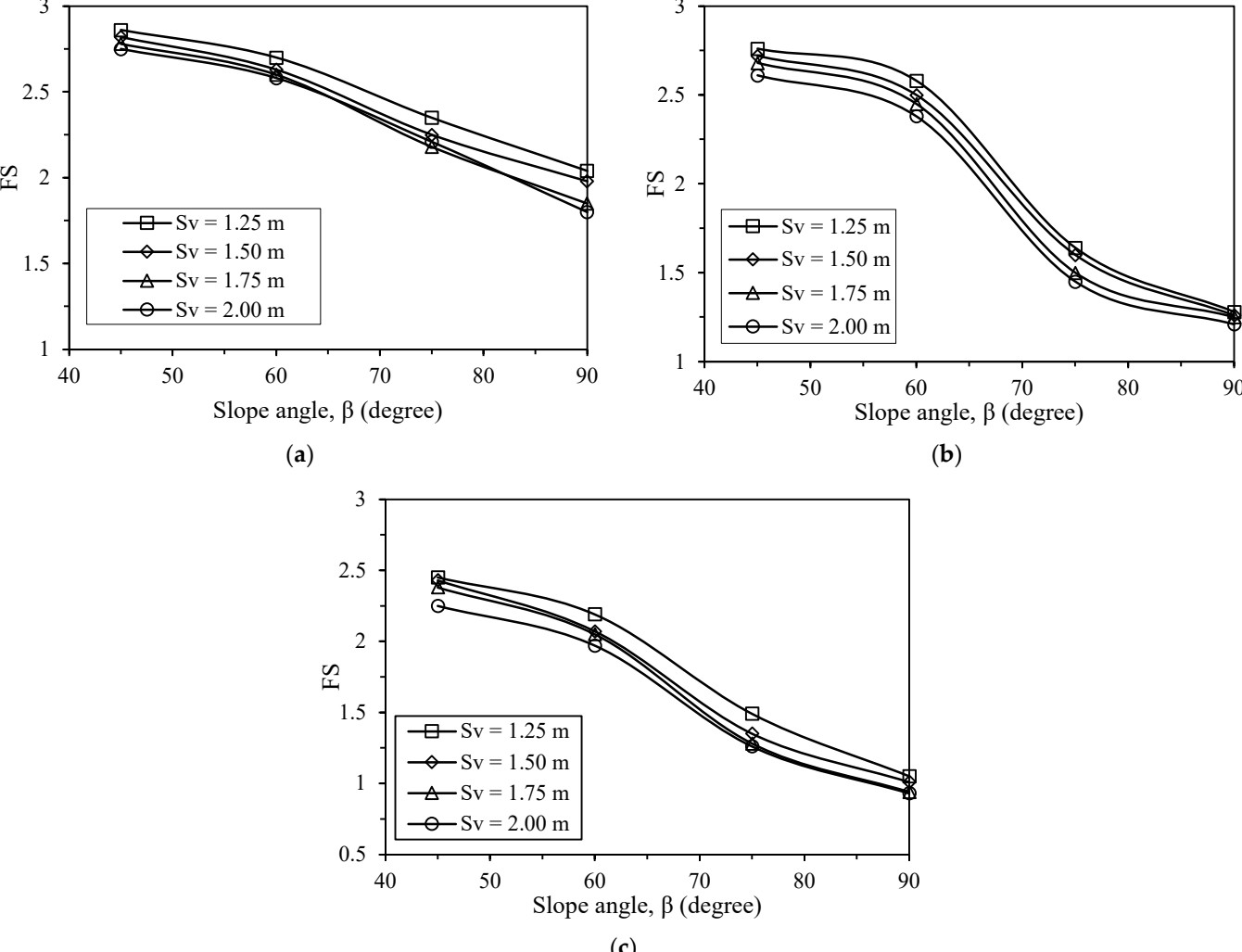

**Figure 9.** Effect of vertical nail spacing on FS of slope with different backslope inclinations: (**a**) $\alpha = 0°$, (**b**) $\alpha = 10°$, and (**c**) $\alpha = 20°$.

### 3.5. Nail Forces

Axial forces coming into nails are dependent on the location and the inclination of soil nails in relation to slip surface. As a result, the contribution of nails located in different positions is different from the overall stability of the slopes [61,62]. Figure 10 shows the distri-bution of axial forces in different nails for various slope and backslope gradients. Nail length (0.9H), inclination (horizontal backslope, optimum nail inclination for $\beta = 45°$, $60°$, $75°$, and $90°$ are $25°$, $20°$, $10°$, and $0°$, respectively. Optimum nail inclinations for $\beta = 45°$, $60°$, $75°$, and $90°$ for $\alpha = 10°$ are $30°$, $25°$, $15°$, and $0°$, respectively; for $\alpha = 20°$, they are $30°$, $30°$, $20°$, and $0°$), and a vertical spacing of 1.5 m was chosen based on the earlier analysis and the fact the horizontal spacing of the nail is 1.0 m. It is observed that with the increase in slope angle, the forces coming into the nail gradually increase from top to bottom. Moreover, as the backslope angle increases from $0°$ to $20°$, the axial force is found to increase, and the maximum force is obtained at the bottom nail for $\beta = 90°$ and $\alpha = 20°$. Bottom nail experiences a tensile force of 15.47 kN for $\beta = 90°$ and for $\beta = 45°$, $60°$, and $75°$; these values are 5.25 kN, 7.83 kN, and 12.43 kN, respectively. The distribution of the nail force for $\beta = 45°$ and $\alpha = 0°$ are shown in Figure 10d. The trend of distribution of the nail is the same for the different nail inclinations. The maximum nail force was found at the bottom nail. This may be due to the irregular distribution of forces along the nails.

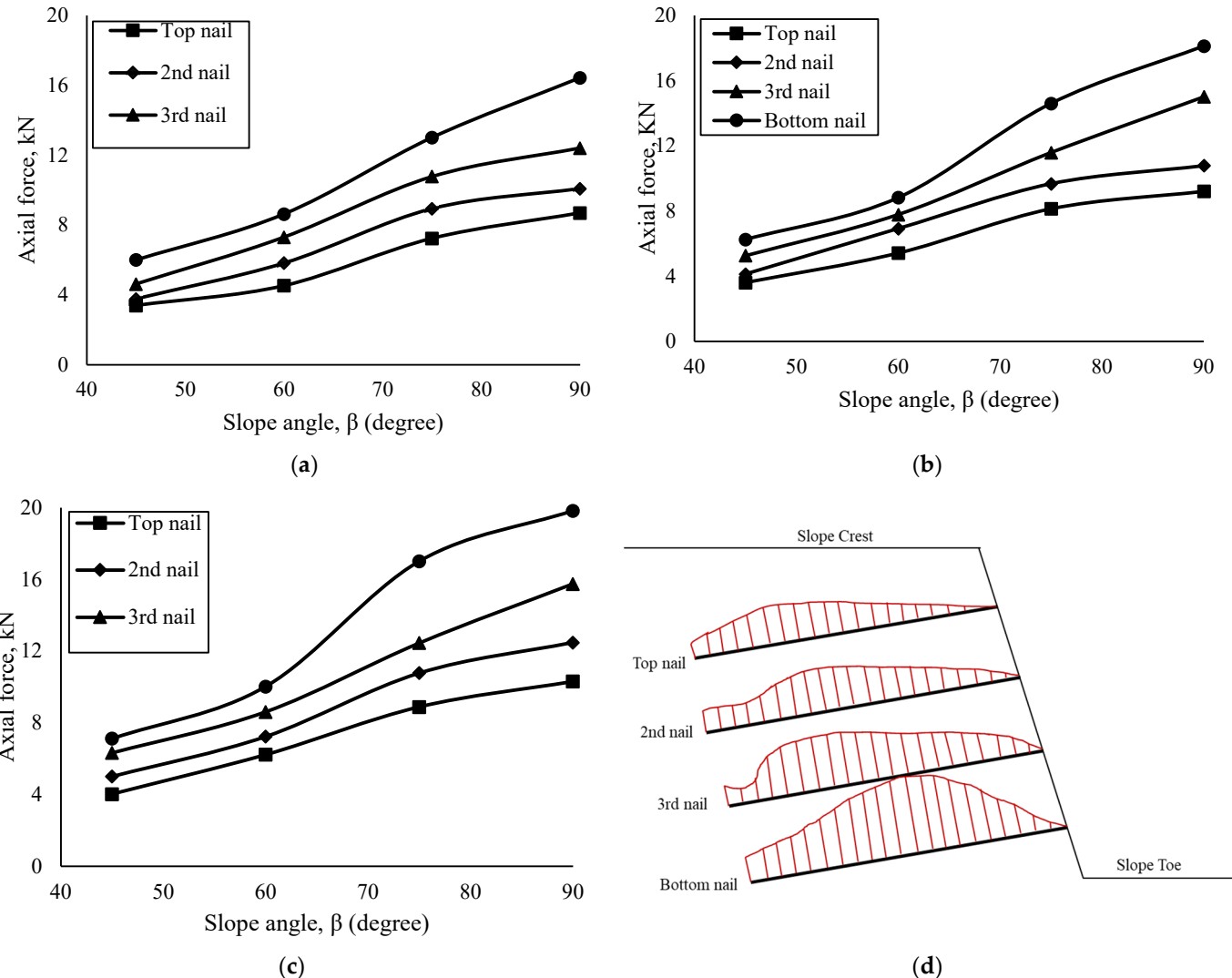

**Figure 10.** Variation of nail forces with slope angles for different backslopes: (**a**) $\alpha = 0°$, (**b**) $\alpha = 10°$, and (**c**) $\alpha = 20°$; (**d**) distribution of nail force for slope angle $60°$ and $\alpha = 0°$.

Rawat and Gupta [45] reported a similar pattern of axial forces coming into the nails as this study; however, the values are different due to having different slope geometries, soils, and nail parameters. Moreover, Figure 10 states that, for β = 45° and 60°, the forces increase gradually for all the nails; however, as the slopes get steeper than 60°, the axial force increases significantly. The force distribution of the nails obtained from the analysis can be a useful guideline while designing the nailing system of the slope. Fan and Luo [1] reported that the nails located at the upper portion of slopes could be made smaller in length as the forces could be mobilized by those nails are less. However, Rotte et al. [7] reported that longer upper nails are effective to restrain the lateral movement of the soil mass. As a result, the length of all the nails used in the study is kept the same for FE analysis.

### 3.6. Influence of Slope Height and Soil Property

The effect of slope height on the stability of the nailed slope is shown in Figure 11a. The factor of safety increases with the increase of slope height and slope angle, which is a common trend for slopes. The effect of the soil properties, i.e., the cohesion and friction angle, is shown in Figure 11b. The higher value of the friction angle and cohesion causes more friction between the nails and soils and thus yields more stability. The factor of safety increased by about 70% when the friction angle increased from 10° to 40°.

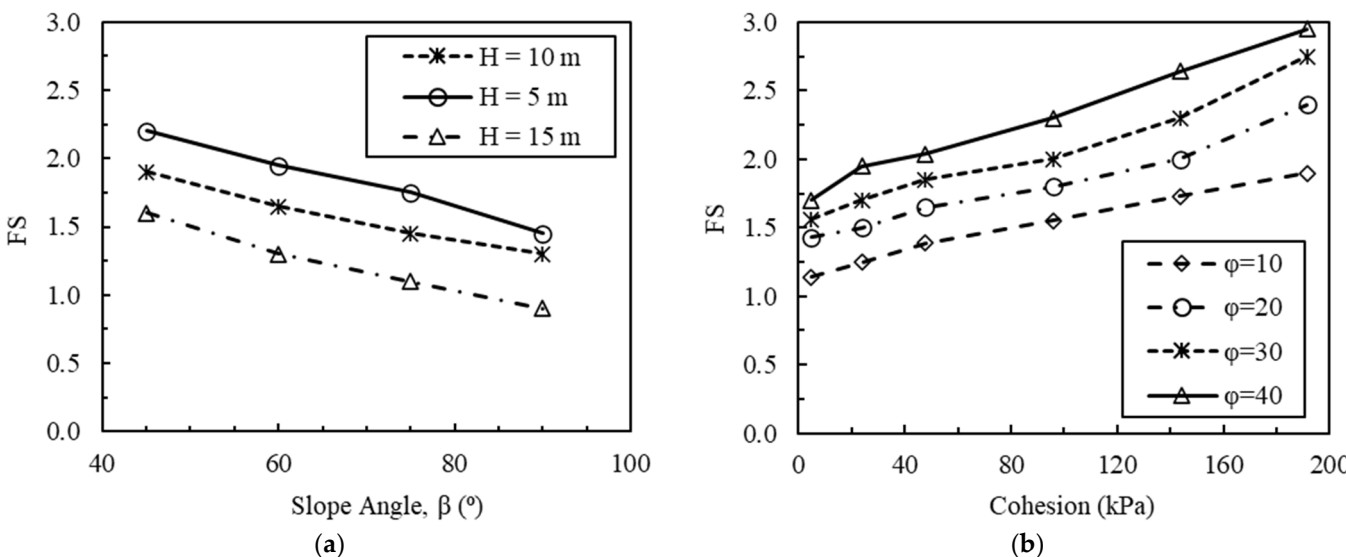

**Figure 11.** Effect of (**a**) slope height and (**b**) cohesion and friction angle on the stability of nailed slope.

### 3.7. Model Comparison and Verification

For verifying the numerical model used in the study, FE analyses are conducted to predict the behavior of soil-nailed cuts, as reported by Mittal and Biswas [63]. Various parameters such as the nail length and inclination, slope angle, cohesion, and friction angle of the soils have been considered for determining the FS of a nailed open cut. Mittal and Biswas [56] performed an analysis of a wall with 6 m height. The nail length varied within 0.5–0.9 times of the height of the wall. Figure 12a presents the variation of FS with the nail inclination for the results reported by Mittal and Biswas [63] and obtained from the FE analysis. It is observed that with the increase in nail inclination, FS increases, and thereafter FS again starts to diminish. A similar pattern has been reported from the results obtained in FE analyses. The results from FE analyses are slightly smaller than those reported by Mittal and Biswas [63]. The difference in values varies within 2.1–11.2% depending on the nail inclination and value of L/H.

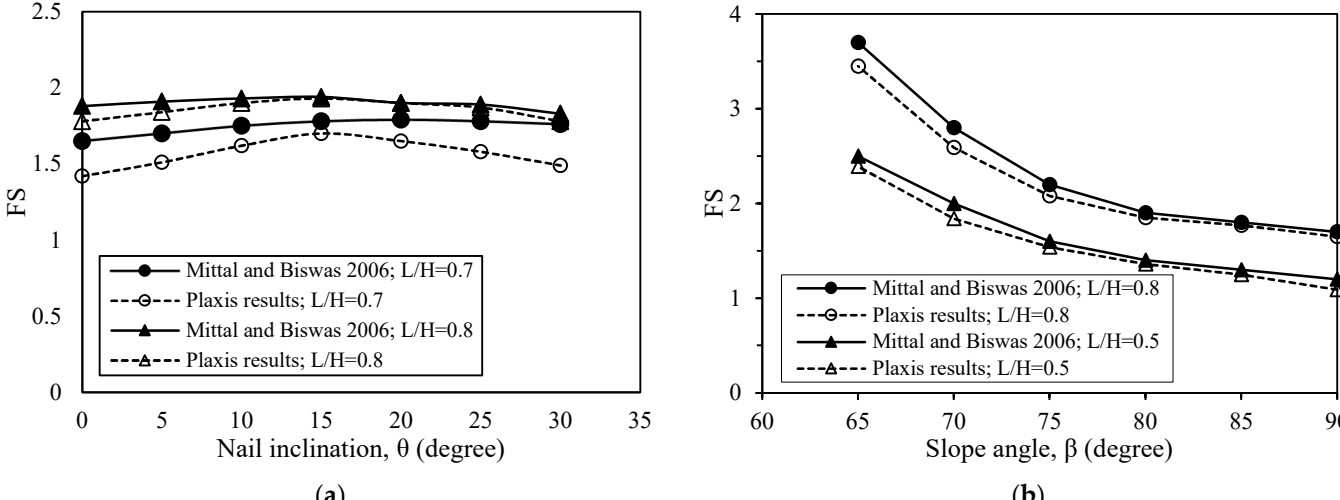

**Figure 12.** Comparison of FS between measured values and results from Mittal and Biswas [55]: (**a**) for different nail inclinations, (**b**) for different slope angles.

In addition, FS for various slope angles with L/H = 0.8 and 0.5 reported in that study are presented with the findings of FE analyses in Figure 12b. FE results vary within 0.5–9.2% of the values reported in their study, and obtained the results are slightly smaller. Moreover, for the verification of the FE model used herein, the FE analysis is performed to predict the maximum nail force of a physical model, as reported by Rawat et al. [2]. The material properties used for the analysis are presented in Table 4. Figure 13 presents the variation of nail force with nail inclination, as reported by Rawat et al. [2] for a slope angle of 45°. The results obtained from the analysis are almost similar to the experimental results reported by Rawat et al. [2], verifying the accuracy of FE analysis considered in the current study.

**Table 4.** Material properties used by Rawat et al. [2].

| Parameter | Value | Unit |
|:---:|:---:|:---:|
| Material model | Mohr–Coulomb | - |
| Type of material behavior | Drained | - |
| Unit weight of soil above the phreatic line | 14.18 | kN/m$^3$ |
| Unit weight of soil below the phreatic line | 18.79 | kN/m$^3$ |
| Young's modulus, E | 50,000 | kN/m$^2$ |
| Cohesion, c | 5.44 | kN/m$^2$ |
| Angle of internal friction, $\varphi$ | 37.0 | ° |
| Outer diameter of nail | 8 | mm |
| Inner diameter of nail | 3 | mm |
| Length of nail | 240 | mm |

The results obtained in the study are also compared with the available existing literature. Figure 14 shows the variation of FS with the nail inclination for the horizontal backslope with β = 60°. FS increases with the increase in nail inclination up to a certain point and thereafter begins to drop. Figure 14 states that the findings in this study agree well with the existing literature. The FS reported by Fan and Luo [1] is higher than the other results, and this is because of the soil used in their study. The soil used for analysis by Fan and Luo [1] had a cohesion parameter, c = 50 kPa and φ = 30°. Although φ is almost similar to the other soil types used, because of the larger cohesion, the FS reported in their study was higher. Rotte et al. [7] reported FS almost similar to the current study since the soil used in their study is similar to the one used for the present study. Although the patterns of the results obtained in the current FE analysis are similar to Fan and Luo [1] and Rotte et al. [3], the differences in values may be due to the different soil types and nail parameters used and the different types of software and their features.

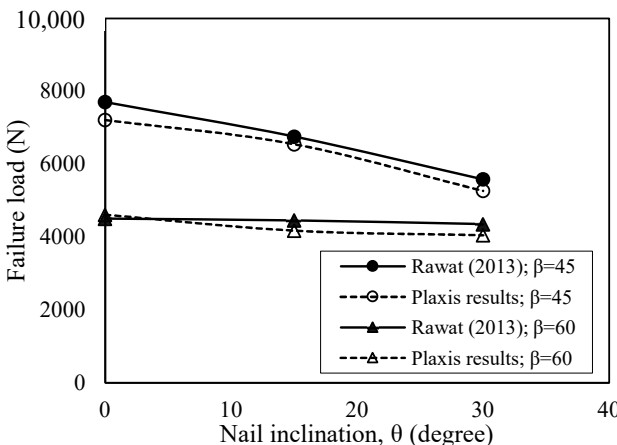

**Figure 13.** Comparison of failure load of a physical model with nail inclination for measured results.

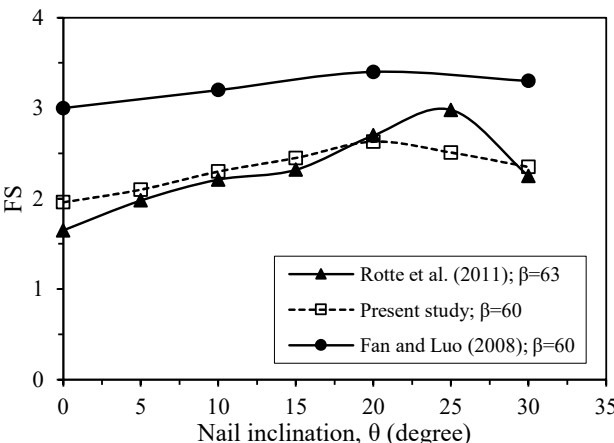

**Figure 14.** Comparison of FS with nail inclination for current study and existing literature.

*3.8. Optimum Nail Layout*

The improved FS of slopes with the optimum nail layout for different types of slope geometry shows that with nailing, FS can be elevated above two, resulting in a stable slope (Figure 15). For those unstable slopes, it can be modified to the slope of 75° and thereafter can be reinforced by soil nail. The inclusion of nailing is found to increase FS by 29–75% depending on slope and backslope angles (Refer to Figure 16), indicating the effectiveness of soil nailing to be used for the stabilization of slopes with sandy soil.

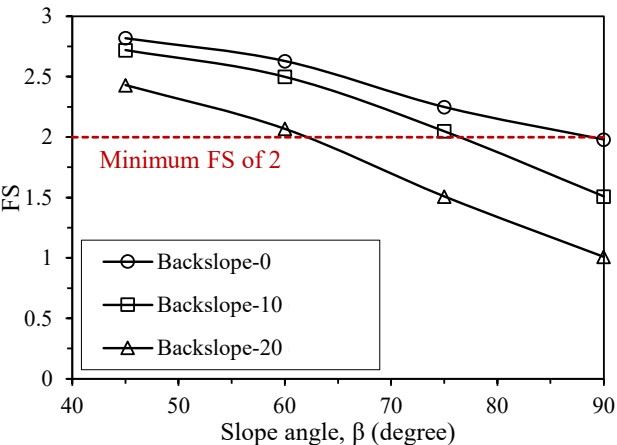

**Figure 15.** Effect of optimum nail layout for enhancing FS with slope angle.

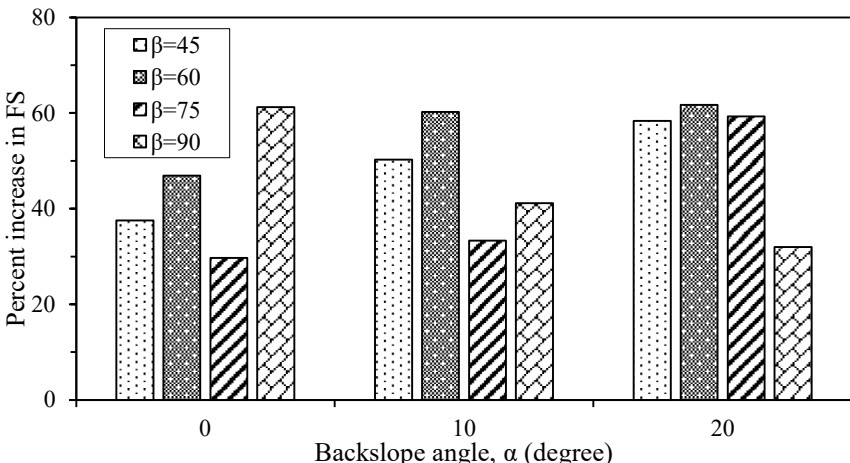

**Figure 16.** Percent increase in FS for various slope geometries due to soil nailing with optimum nail parameters.

*3.9. Wall Facing*

Soil nails should be connected to a facing system at the excavation face or slope surface. The combination of soil nails and facing should be dimensioned to sustain the expected maximum destabilizing force. There are different facing types that are available for a nailed slope system. Shotcrete facing is typically less costly than the structural facing required for other wall systems [64]. A bioengineering technique using vegetation can be used as a substitute for structural facing, which is effective for controlling the erosion and sloughing of the soil at the slope surface [10,65–67]. However, the use of this technique should be limited to non-critical structures where large vertical deformation and horizontal deformation are acceptable.

**4. Conclusions**

This study has been carried out to find the optimum nail layout for stabilizing the slopes effectively. The effect of the nail parameters—nail inclination, length, spacing, and forces—on slope stability has been investigated through FE analysis. The effectiveness of soil nailing has been studied, and an optimum design nail layout is found out for stabilization. Based on the analyses carried out in the study, the following conclusions can be drawn:

- For a horizontal backslope, the optimum nail inclinations for $\beta$ = 45°, 60°, 75°, and 90° are 25°, 20°, 10–15°, and 0°, respectively. For a backslope of 10°, the optimum nail inclinations for $\beta$ = 45°, 60°, 75°, and 90° are 30°, 25°, and 15–20°, whereas for a backslope with 20°, the optimum nail inclinations for $\beta$ = 45°, 60°, 75°, and 90° are 30°, 30°, 20°, and 0° respectively. Moreover, the importance of selecting an optimum nail inclination is understood through failure surface observation from FE analysis.
- With the increase in the value of L/H, FS is found to increase for all the slope angles, and the percent increase in FS varies within 23–63% depending on L/H and the slope angle. FS increases rapidly with the increase in values of L/H up to 0.9, and beyond that the increase in nail length increases FS by only 1.7–4.3%. Therefore, the optimum nail length is chosen as 0.9 times the height of the slope.
- With the decrease in the vertical spacing of the nail, FS is found to increase, and the maximum FS is found for a vertical spacing of 1.25 m. Although reduced vertical spacing improves the FS of slopes, the difference in FS for the spacing of 1.25 m and 2.0 m varies within 9–17%. Considering the economy and ease of construction, a vertical nail spacing of 1.50 m can be selected as the optimum nail vertical spacing.
- The axial force entering the nail increases with the increase in slope and backslope angle. For all the cases, the bottom nail is found to experience a maximum tensile force, whereas the minimum tensile force is observed in the topmost nail.

- An optimum nail layout has been proposed for similar soil types and slope geometries. The inclusion of soil nails with the optimum layout can increase FS by 29–75%.

The present study analyzed the effect of various nail parameters on stabilizing the hill slopes and suggested an optimum combination of these parameters to stabilize the existing vulnerable hill slopes. Along with the numerical study, the development of a physical model to simulate the behavior of soil nails would be useful to warrant the use of soil nailing in stabilizing slopes. Moreover, future studies could consider a life cycle analysis to assess the feasibility of using soil nails, and they might consider that economic and comparative analysis could be done with the other existing methods of slope protection.

**Author Contributions:** Conceptualization, T.E.E. and M.S.I.; methodology, T.E.E. and M.A.I.; software, T.E.E. and M.A.I.; validation, T.E.E. and M.A.I.; formal analysis and investigation, T.E.E. and M.S.I.; writing—original draft preparation, T.E.E.; writing—review and editing, M.A.I. and M.S.I.; and supervision, M.S.I. All authors have read and agreed to the published version of the manuscript.

**Funding:** This research received no external funding.

**Institutional Review Board Statement:** Not applicable.

**Informed Consent Statement:** Not applicable.

**Data Availability Statement:** The primary data, models, and code generated or presented in this study appear within the article. The detailed data are available on request from the corresponding author.

**Acknowledgments:** The authors acknowledge the infrastructural and financial support received from Bangladesh University of Engineering and Technology (BUET), Dhaka, Bangladesh for carrying out the research work. The authors would also like to thank BUET-Japan Institute of Disaster Prevention and Urban Safety (BUET-JIDPUS) for technical assistance regarding PLAXIS.

**Conflicts of Interest:** The authors declare no conflict of interest.

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
