# Peer review of "Parametric Assessment of Soil Nailing on the Stability of Slopes Using Numerical Approach"

_geotechnics, doi:10.3390/geotechnics2030030_

Round 1
Reviewer 1 Report
The reviewer comments are given in the annotated manuscript pdf.

Author Response
REVIEWER 1:
|
Reviewer’s comments |
Authors’ Response |
|
Please refer to FHWA manual on soil nailing. This statement is not correct.
|
Authors agree with the reviewer and FHWA provides a general guideline regarding the suitable soil type for soil nailing. Therefore, the statement is omitted. |
|
Change to "significant loss of human life" |
Changed |
|
Please mention that work is exclusively for Rangamati district as similar work on optimum soil nail layout and parametric variation is available in abundance in literature. |
Thank you for the comment. In the revised manuscript it is mentioned that the analysis is carried out for Rangamati of Bangladesh to propose an optimum nail layout system. Please refer to line 115-118. |
|
How were the soil sampling done? Please mention the codal provisions followed. How was the depth and number of sampling decided?
|
Soil samples have been collected from the hill slopes with the help of Shelby tube and after collection, the tubes were waxed at both ends. Depth of collected soil samples was 1.0-2.0 m. In total, five Shelby tube samples have been collected and all the soil had more or less similar characteristics in terms of geotechnical properties. Therefore, only one soil sample was considered for the analysis. ASTM D1587-08 code has been followed for thin-walled tube sampling of soils for geotechnical purposes. |
|
Provide a legend/slab below the abscissa. |
Legend has been provided in Figure 1. |
|
What about the calibration of the FE model? The authors have not mentioned anything regarding the scaling of the model? The boundary effects, scaling of materials is missing. This is essentially required to be mentioned? Did the authors used the same dimensions of the slope as at the site? Please include a schematic or actual image of the slope with dimensions for better understanding of the readers. Also explain how the dimensioning has been done in the FE environment. |
The boundary conditions are discussed in Line 176-180. Authors have observed the height of the existing hill slopes in the site to be varied within 10-15 m and most of the slopes were around 10m as such height of 10m was considered in the analysis. Therefore, no scaling was done. The authors also conducted few more analysis to observe the effect of slope height on the stability of nailed slope. The results are presented in Figure 10. A schematic diagram of the slope with dimension and a FEM model of the analyzed slope is provided in Figure 2(b) including the meshing and boundary conditions for better understanding of the reader. Thanks.
|
|
The authors should justify the bar diameter and grout diameter. Why has the authors used only 10 m when the available height of slope is 10 - 15m? It would have been better to take the avg. ht of the maximum ht.
|
The authors have used here #9 rebar (29 mm in diameter) which is commonly available in the local market, and a typical grout diameter of 10 cm is used which is also used in existing literature (Fan and Luo 2008; Singh and Babu 2010). Authors have observed the height of the existing hill slopes in the site to be varied within 10-15 m and most of the slopes were around 10m as such height of 10m was considered in the analysis. |
|
what was the actual slope angle at the site for which sampling was done? Is there any specific reason to use the slope angles between 45 - 90?
|
The actual slope angle is found to vary within 30-75o in the site. However, literature (Elahi et al. 2018; Elahi et al. 2019) reported that at times hill slopes are cut to very steep slope almost 80-90o in these hilly areas of Bangladesh. As such authors considered the slope angles within 45-90o. |
|
How did the authors choose the variation between 5 - 12 m as the length is the function of the unsupported wall height? Explain. |
Authors wanted to find the appropriate length of the nail with respect to the height of the slope as it has relationship with the location of failure plane. Therefore to optimize the length of the nail in practical range, authors considered the L/H values of 0.5-1.2. |
|
Please mention all the dimensions of the slope? |
Please refer to Figure 2 for all the dimension of slopes. |
|
What about the other boundary conditions? elaborate |
Please refer to Line No 176-180 for detailed boundary condition. The boundary conditions are also shown in Figure 2(b). |
|
How did the authors determine the dilatancy? |
Authors have taken the value of dilatancy angle based on the literature where MC model were adopted for the FE analysis (Rawat and Gupta 2016; Rawat et al. 2016; Singh and Babu 2010). |
|
Why did the authors considered drained soil condition when the soil is SC? |
The authors considered drained soil behavior since sandy soil has high permeability and the excess pore water pressure quickly dissipates. |
|
Please give the reason for using the virtual thickness as 0.1. |
Authors thank the reviewer for the valuable comment. This virtual thickness factor of 0.1 is used to ensure proper soil-nail interaction and this value is taken based on the literature. Sharma et al. (2018), Rawat and Gupta (2016) have used this value to analyze the stability of nailed slope accurately. Please refer to line 208-212. |
|
What laboratory tests are the authors referring to for determination of R inter value? How and why did the authors select R inter = 1? |
Chu and Yin (2005) performed series of laboratory pullout tests and interface shear tests on grouted nail and soil to determine the value of interface strength and found the value to exist within 0.95-1.07 and based on the results of those experiments, Rinter value was considered 1 in the study. Similar value is used for the FEM analysis by Fan (2008) and Rawat and Gupta (2016) for slopes stabilized with soil nail. Please refer to line 208-212. |
|
The authors should show the mesh sensitivity analysis? IS the entire slope has been meshed fine? If yes then explain why? |
Please refer to Figure 2 for the FE model used where the meshing of the model is shown. The finer mesh will increase the accuracy of the obtained results compared to coarser mesh and as such it was used in the study. |
|
The final FE model in Plaxis 2D should be given with all the markings? |
Please refer to Figure 2 (b) for detailed model used in Plaxis 2D. |
|
What about the loading? How was the loading applied? The authors should describe the loading on the slope? |
Authors thank the reviewer for this important point. No external loading was applied for the studied model rather the weight of the unreinforced slope and nailed slope was considered for the calculation. In this particular case in Rangamati, no surcharge load was found during the field observation. Therefore, to represent the actual field condition no external load was considered. |
|
Please use the word 'unreinforced'
|
Changed to unreinforced. |
|
The authors should refer to https://www.cedd.gov.hk/eng/publications/geo/geo-reports/geo_rpt197/index.html. The claim is not true because for the same reason longer nails at the top and smaller nail lengths towards the slope toe is generally recommended for making the nail intersect the slip surface. The nail inclination variation is generally from 0 - 15 deg. for horizontal backfill slope angle. The authors should check the tensile strain development and the nail forces along the nail length for more accurate explanation. This is very significant for discussing the nail inclination.
|
Authors thank the reviewer for this valuable point. Authors agree with the reviewer regarding the importance of nail inclination to provide shorter nail length at bottom to intersect with the slip surface. The authors analyzed the pattern of nail force distribution and it was observed that nail force is found to increase from the end and reach a maximum value then again tend to decrease near the nail head. The mobilization of axial forces also depend on location and inclination of nail with respect to slip surface. The nail length beyond slip surface in the upper nails is small and can be considered less significant compared to the bottom nails but it also plays an significant role in reducing the lateral movement of nailed slope as such the length of the nails has been kept same throughout the slope as well. On the contrary, maximum tensile force mobilized in individual nail decrease gradually with the increase in inclination which is also the reason that optimum FS for the presented case in Figure 5 is for nail inclination of 20o and decrease with the further increase in inclination. |
|
The authors should show for different backfill slope angles.
|
The backfill angle of 10° is shown in Figure 2(b). In figure 5, backfill angle of 0° is shown. |
|
What type of displacement are the authors referring to when stating 'top displacement', horizontal or vertical? If vertical then it is suggested to use crest settlement. If the authors are referring to horizontal displacement, then explain how are the values determined from numerical analysis.
|
Authors have considered vertical displacement here in the study and as such revised the term to crest as suggested by the reviewer. |
|
How are the variations from 1.25 to 2.0 selected? FHWA suggested one nail for 4m^2 area for field application? Explain. The effect should be considered for both vertical and horizontal spacing? Why is the effect of horizontal spacing neglected? Is 1.5 m recommended for horizontal spacing also. Please discuss. |
The variation of vertical and horizontal spacing was considered based on the studies performed by Rawat et al. (2013), Rawat and Gupta (2016), Singh and Babu (2010) for the similar nail and grout arrangement considered in the study. Although reduction in vertical spacing improved the FS but the increase was in very narrow range. And 1.5 m to 2.0 m spacing is commonly used in the existing studies as such authors have selected 1.5 m. |
|
Nail force distribution along each nail from the Plaxis Output is required to be presented here. It is also essential for understanding and discussing the effect of nail inclination. |
Nail force distribution of each nail has been shown in Figure 9(d). |
|
When the major soil mass is moving at the top during failure, how is it possible that minimum tensile forces are mobilized along the nail in the top rows? Please justify |
If the top nail doesn’t intersect the slip surface, then there will be no mobilization of forces at the top nails (for example, Figure 5a case). Therefore, the maximum force can be found at the bottom nail. However, depending on the failure surface, this scenario can change. |

Reviewer 2 Report
In this manuscript, FEM was adopted to investigate the impacts of different influential factors of soil nailing and slope on the slope stability. In general, the parametric study is comprehensive, most of primary parameters have been discussed. However, the PLAXIS is a wide-used commercial software, leading a relatively poor novelty. Therefore, the content of this manuscript should be further perfected for publication. After reviewing this manuscript, I believe that the following issues need to be discussed, and there are grammar and format problems that need to be revised as well.
The author stated that the slope height varies within 10-15m in their selected field therefore, they chose slope height equals to 10m in this study. However, it is well-know that the slope height is various in engineering practice. If this study only considered cases of H=10m, the conclusion is too limited, which cannot provide enough guidance to engineering practice. In my view, H=5m and 15m should be considered briefly. In addition, the influence of soil properties also should be investigated. Actually, in Table 1, only the cohesion and the internal friction can significantly affect the results. I suggested that these two factors should be considered. The authors don’t need to added too much analysis. For example, adding two more cohesion and internal angle in Fig. 6(b) only requires eight more numerical simulation, which is totally affordable. Furthermore, a sketch including the information of the arrangement of meshes, the boundary conditions and the number of elements should be presented for a better readability. And the background introduction is not comprehensive enough, especially for the absence of recent articles regarding the slope stability. Following recent studies may helpful for optimizing this manuscript:
Undrained Bearing Capacity of Strip Footings Placed Adjacent to Two-Layered Slopes;
Undrained seismic bearing capacity of strip footings lying on two-layered slopes;
Undrained seismic bearing capacity of strip footings horizontally embedded in two-layered slopes;
Undrained stability analysis of eccentrically loaded strip footing lying on layered slope by finite element limit analysis;
Ultimate Bearing Capacity of Strip Footings on Hoek–Brown Rock Slopes Using Adaptive Finite Element Limit Analysis.
Author Response
REVIEWER 2:
In this manuscript, FEM was adopted to investigate the impacts of different influential factors of soil nailing and slope on the slope stability. In general, the parametric study is comprehensive, most of primary parameters have been discussed. However, the PLAXIS is a wide-used commercial software, leading a relatively poor novelty. Therefore, the content of this manuscript should be further perfected for publication. After reviewing this manuscript, I believe that the following issues need to be discussed, and there are grammar and format problems that need to be revised as well.
|
Reviewer’s comments |
Authors’ Response |
|
1. The author stated that the slope height varies within 10-15m in their selected field therefore, they chose slope height equals to 10m in this study. However, it is well-know that the slope height is various in engineering practice. If this study only considered cases of H=10m, the conclusion is too limited, which cannot provide enough guidance to engineering practice. In my view, H=5m and 15m should be considered briefly. In addition, the influence of soil properties also should be investigated. Actually, in Table 1, only the cohesion and the internal friction can significantly affect the results. I suggested that these two factors should be considered. The authors don’t need to added too much analysis. For example, adding two more cohesion and internal angle in Fig. 6(b) only requires eight more numerical simulation, which is totally affordable |
The effect of slope height with different slope angles is shown in Figure 10 (a). Also, the effect of friction angle and cohesion is shown in Figure 10 (b). |
|
2. Furthermore, a sketch including the information of the arrangement of meshes, the boundary conditions and the number of elements should be presented for a better readability. |
Authors thank the reviewer for this important point. Please refer to Figure 2 (b) which shows the detail of FE model and authors hope it would increase the readability of the paper. |
|
3. And the background introduction is not comprehensive enough, especially for the absence of recent articles regarding the slope stability. Following recent studies may helpful for optimizing this manuscript: Undrained Bearing Capacity of Strip Footings Placed Adjacent to Two-Layered Slopes; Undrained seismic bearing capacity of strip footings lying on two-layered slopes; Undrained seismic bearing capacity of strip footings horizontally embedded in two-layered slopes; Undrained stability analysis of eccentrically loaded strip footing lying on layered slope by finite element limit analysis; Ultimate Bearing Capacity of Strip Footings on Hoek–Brown Rock Slopes Using Adaptive Finite Element Limit Analysis.
|
Authors are grateful to the reviewer for suggesting these useful papers to revise the manuscript which are included in this updated manuscript. Please refer to Reference 22-26 of this revised manuscript. |

Author Response
REVIEWER 3:
|
Reviewer’s comments |
Authors’ Response |
|
The authors shall make clear in the introduction, and generally throughout the paper, what is the novelty and the original parts of their work, and what is the literature and knowledge gap that they try to cover. Many similar works (parametric analyses of soil nail walls) have been presented in the past (for instance Fan and Luo, 2008, on which the authors often refer); what is the impact of the present work and where exactly does it add to the knowledge base? It was not very clear to the reviewer, and the authors should try to clarify these issue.
|
Authors thank the reviewer for these important points. The present study is attempted to propose an optimum soil nail layout considering various nail parameters that can be used to protect the hill slopes of Chattogram hill tracts of Bangladesh especially, Rangamati. In the existing literature, different methods including vegetation have been studied to stabilize those slopes, however, a complete parametric study considering soil type, backslope angle and existing slope angles and heights of Rangamati is missing. Therefore, the study was conducted to provide a useful guideline to protect the existing hill slopes with soil nailing to reduce the significant loss of life and damaged caused by landslides in this region. Please refer to Line 107-126. |
|
A lot of information related to the numerical model is missing and this creates a lot of questions. Information on issues such as the overall geometry of the model (dimensions), the generated mesh (how fine or coarse, how many elements), the densification or not near the critical areas (around the nails), the interface issues near the rear end of the nails, the ignorance or not in modeling the grout around the nails is missing.
|
Authors apologize for not clearing the detail of numerical model in the earlier manuscript. Authors have updated the Table 2 showing the different parameter of MC model, grout properties, provided a new figure (Figure 2b) indicating the details of FE model, described the boundary condition (Line 176-180) and consideration of other parameters (Line: 208-212; 2016-220) for the FE analysis. Also, grouting is considered around the nail in the models (Line 149-151). The properties of the grout has been included in Table 2. |
|
In several occasions the manuscript becomes very wordy (while in other occasions necessary information is missing). |
Authors have omitted and modified a lot of portions of the paper as suggested by the reviewer and authors believe that this updated version is free from such verbosity. |
|
Introduction |
|
|
- Completely omit lines 45-60. - Line 32: Include some more fundamental references about soil nails (in addition to 1, 2). - References 13-14 are irrelevant to the subject of the paper. Omit them. - Lines 73-103: It would be good to categorize all the relevant references based on whether the study was experimental, numerically-based, limit equilibrium based. - Lines 112-123: Irrelevant to the purpose and the subject of the paper. Completely omit these lines. - Lines 125-135: Clearly state what is the gap in the literature, and what is the new, innovative part that your study introduces.
|
- Mostly omitted and modified.
- New references are added.
- References 13-14 are omitted
-Authors have mentioned the type of analysis in the manuscript for all the references.
-Lines have been omitted mostly and remaining lines have been shortened as per the suggestion of the reviewers.
- The main objective of this study was to propose an optimum nail layout for different slope conditions considering slope and backslope angle (which is rarely reported in literature) to protect the vulnerable hill slopes of Chattogram Hill Tracts of Bangladesh. Various studies have been conducted to suggest the reason of slope failure in this zone and various methods have been suggested, however, such parametric study considering the different parameters of nails is missing in literature. Therefore, the proposed optimum nail layout could be a useful guideline for implementing soil nailing to prevent slope failures in the country. Please refer to line 107-126. |
|
Materials and Methods |
|
|
- Omit Table 1 and Figure 1; they offer nothing to the purpose of your work. Instead, present all the details about the Mohr Coulomb parameters (dilatancy angle, Poisson’s ratio which are now missing). - Omit lines 158-159. - Elaborate on the following: dimensions of the geometry, densification of the mesh (particularly near the nails and the nails), boundary conditions, interface at the rear end of the nails, virtual thickness (why 0.1?), simulation of grout properties. - Table 2: What about grout properties - Clearly refer to and present the baseline model.
- What is the matrix (density) of the nails in the horizontal and vertical direction?
|
- The mentioned properties are included in Table 2. Dilatancy angle, Poisson’s ratio, modulus of elasticity of nail and grout are added in Table 2.
- Omitted
- Please refer to figure 2 for the dimension of the geometry and meshing of the model. Boundary conditions are elaborated in Line 176-180. Discussion on virtual thickness has been addressed in Line 208-212. Grout diameter and modulus of elasticity are mentioned in Line 149-151 and Table 2 respectively.
- The vertical spacing of nail has been varied within 1.25-2.0 m and horizontal spacing is maintained at 1.0 m. |
|
Results and discussion |
|
|
- The whole section 3.1 (Stability analysis of bare slopes) shall be omitted. - Omit lines 255-257 (from When soil… until mechanism [1, 54].). - Lines 295-296: In order to help the reader, rather than forwarding him to Figure 4, do mention which are the (optimum) inclinations that you used in the models of Figure 6a, 6b, 6c. - Lines 314-315: Here it gets a bit confusing: until now you have been working with phi-c reduction to obtain the SFs, and suddenly you refer to displacements. Is this (Figure 7) the result of a phi-c reduction analysis or of a plastic analysis?
|
- Shortened mostly.
- Omitted
- Optimum nail inclination that were used is stated in the line 294-297.
- For measuring the displacement, plastic analysis was performed rather than phi-c reduction. Authors apologize for not mentioning it in the earlier manuscript. |
|
Section 3.4 |
|
|
- Lines 325-326: clearly mention here which one is the optimum combination on which you refer. - Section 3.4: It is not clear whether the number of nails remains constant, i.e. it is not clear what happens with the out-of-plane density of soil nails. You shall clarify. Does the out-of-plane density remain constant? - Lines 331-333: what do you mean by “… when the number of nails used in slope is unchanged.”? - Lines 335-337: The statement “Considering the effectiveness, economy and analyzing the obtained FS presented in Figure 8, nail spacing of 1.5 m can be considered as optimum vertical nail spacing.” is vague and the conclusion (that the 1.5 m spacing is the optimum) is not supported by arguments. Why not 1.25, or why not 1.75 or 2.0? The differences between all four cases are, as the authors state, rather small. Section 3.5 |
- Mentioned.
- In the analysis, vertical spacing of the nail is varied whereas horizontal spacing was kept 1.0 m. That means, whatever the vertical spacing of nail is-such as 1.25m, 1.50m, 1.75m and 2.0m, horizontal spacing is 1.0m. - Fan and Luo (2008) reported that if number of nail is kept constant and only the vertical spacing among them is changed then the effect is negligible. The statement is rephrased for better understanding.
- The statement is corrected and mentioned that any vertical spacing of 1.25-2.00 m can be chosen for providing nails. It is also stated that for the further analysis in the study, spacing of 1.5 m was considered. |
|
- Lines 342-345: Same comment as before - clearly mention here which one is the optimum combination on which you refer on this section. For instance, make clear in particular with respect to Figure 9, how many vertical layers of nails do exist in these analyses, as well as what is the horizontal distance between nails. - Figure 9: Rather than simply showing the axial forces (as obtained by Plaxis), it would have been better to present the SFs against tensile and pull out failure per soil nail layer (or per soil nail).
|
- Please refer to Line 346-350 and suggested corrections are done.
-Authors acknowledge that the factor of safety against tensile and pull out failure per soil nail layer would have been better for the presentation. To understand this behavior, nail force distribution diagram has been added (Figure 9d). However, authors did not investigate this broadly as it will lengthen the manuscript. |
|
Section 3.6 |
|
|
- This section has been confusing to the reviewer and its title has been misleading. The fundamental comment is that both the Mittal and Bidwas (2006) and the Fan and Luo (2008) papers are also theoretical studies, and not real case studies for which in-situ measurements and results were presented. So, the authors cannot refer to these two papers for “validation” purposes, but just for comparison purposes. The term validation refers to the comparison of a (numerical or analytical) model with a large (or at least laboratory) scale model for which data are available. So, the title of the section is misleading and the word “validation” has to be modified. - As far as the Rawat et al. (2013) study is concerned, this only refers to failure loads of nails, but it is not clear at all what is it that the authors compare with. Did the author create a numerical model of what Rawat et al. did, and then compare their results? Totally vague and unclear. - The Mittal and Bidwas (2006) paper refers to a 6m high wall: do the authors compare the results with their 10m high wall? Please, clarify. - Figure 11: The axial forces of Figure 11 are two (!) orders of magnitude larger than the results shown in Figure 9!
|
- The title has been modified and validation term is omitted. Authors completely agree with the comment of the reviewer and express their sorrow for not considering it in the earlier version of manuscript.
- Rawat et al. (2013) performed both experimental and FE analysis of a nailed slope and determined the nail force vs settlement curves. And both the results were in close agreement with each other. In this section, authors have selected the results of their study for a similar slope angles of 45o and 60o for comparison for similar nail orientations.
- Regarding the study of Mittal and Biswas (2006), length was different with the current study, however, authors have attempted to compare the results of both the studies in terms of L/H ratios.
- There was a mistake in plotting in the Figure 9 and all the values in y-axis were 10 times higher which is corrected in this manuscript. Authors are sorry for this mistake and hope the reviewer will consider it. |

Reviewer 4 Report
1. Abstract needs revision, very obvious results are mentioned. Incorporate significant outcomes from the study in the abstract.
2. Authors have performed an appreciable literature review, however, following studies can be mentioned to enhance the impact of the article:
https://doi.org/10.1007/978-981-16-1993-9_11
https://doi.org/10.1080/19648189.2013.828658
https://doi.org/10.1007/978-981-15-9984-2_19
https://doi.org/10.1016/j.sandf.2014.11.011
3. line 172: It assumes infinite length perpendicular to plane section, and out of the plane displacement is zero. The statement is incorrect, the strain in the largest dimension is zero, authors must revise suitably. They may refer following references for more clarification on plane strain conditions:
https://doi.org/10.1007/s12517-022-10366-1
https://doi.org/10.1007/s40891-021-00325-3
4. Provide proper explanation for using 15 noded triangular elements. No suitable explanation of vertical boundary conditions is present in the paper.
5. The modulus of elasticity of grout material is based upon experiment performed by the authors? or taken randomly, mention proper reference for the same
6. Why interface parameter is considered as 1 here in this case, is it not too high?
7. "A virtual thickness was assigned to the interface element in order to define the material characteristics. A virtual thickness factor of 0.1 is used in the analysis." How did the authors decide the thickness factor? How it is simulated? Show mesh figure for the same.
8. What does the slip surface indicate? why did the vertical spacing between nails are varied in a single numerical model (Fig 5)?
9. Conclusion section needs revision, kindly present the outcomes more clearly.
Author Response
REVIEWER 4:
|
Reviewer’s comments |
Authors’ Response |
|
1. Abstract needs revision, very obvious results are mentioned. Incorporate significant outcomes from the study in the abstract. |
Abstract has been revised to reflect the significant findings of the study. |
|
2. Authors have performed an appreciable literature review, however, following studies can be mentioned to enhance the impact of the article: https://doi.org/10.1007/978-981-16-1993-9_11 |
Authors thank the reviewer for suggesting these important studies and authors have included those in this revised manuscript. Please refer to reference 3-6. |
|
3. line 172: It assumes infinite length perpendicular to plane section, and out of the plane displacement is zero. The statement is incorrect, the strain in the largest dimension is zero, authors must revise suitably. They may refer following references for more clarification on plane strain conditions: https://doi.org/10.1007/s12517-022-10366-1 |
Thank you for the important point. Authors have modified the sentences (166-168) and revised it based on the references provided by the Reviewer. |
|
4. Provide proper explanation for using 15 noded triangular elements. No suitable explanation of vertical boundary conditions is present in the paper. |
Authors are grateful to the reviewer for raising this important point. 15-noded triangular elements are effective to estimate the results more accurately compared to six-noded element and it provides fourth order interpolation for displacement and integration. This method is accurately employed in different literature to predict failure load and factor of safety in phi-c reduction method (Mohamed 2010; Rawat and Gupta 2016; Fan and Luo 2008). These discussions are included in this version of revised manuscript. Please refer to Line 169-176. Please refer to the lines 176-180 about the boundary conditions of the of the used model. |
|
5. The modulus of elasticity of grout material is based upon experiment performed by the authors? or taken randomly, mention proper reference for the same |
Authors express their sorrow for not clarifying it in the earlier manuscript. The modulus of elasticity of grout material used in the study is based on the study conducted by Singh and Babu (2010) and the value is 22 GPa which is a typical value for concrete grout. Authors have added the value also in the Table 2 for readers to understand. |
|
6. Why interface parameter is considered as 1 here in this case, is it not too high? |
Chu and Yin (2005a) and Chu and Yin (2005b) performed series of laboratory pullout tests and interface shear tests on grouted nail and soil to determine the value of interface strength and found the value to exist within 0.95-1.07 and based on the results of those experiments, Rinter value was considered 1 in the study. Similar value is used for the FEM analysis by Fan (2008) and Rawat and Gupta (2016) for slopes stabilized with soil nail. Please refer to line 216-220. |
|
7. "A virtual thickness was assigned to the interface element in order to define the material characteristics. A virtual thickness factor of 0.1 is used in the analysis." How did the authors decide the thickness factor? How is it simulated? Show mesh figure for the same. |
Authors thank the reviewer for the valuable comment. This virtual thickness factor of 0.1 is used to ensure proper soil-nail interaction and this value is taken based on the literature. Sharma et al. (2018), Rawat and Gupta (2016) have used this value to analyze the stability of nailed slope accurately. Please refer to line 208-212. The mesh figure is shown in Figure 2b. |
|
8. What does the slip surface indicate? why did the vertical spacing between nails are varied in a single numerical model (Fig 5)? |
Slip surface is generally defined as the surface along which the slope is likely to fail and in Figure 5, slip surface for different nail orientations are demonstrated. The vertical spacings between the nails were kept same to investigate the influence of single parameter. |
|
9. Conclusion section needs revision, kindly present the outcomes more clearly. |
Conclusion has been revised to reflect the outcomes more clearly. |

Reviewer 5 Report
Although the topic is interesting, nevertheless the paper does not present a high level of innovation. The content is more about applying a well-known stabilizing method to a case study slope, including a parametric numerical analysis, than actual research.
General comments:
1) Table 1 needs amendments;
2) It is not clear which is the model assumed at the interface between the nail and the surrounding soil;
3) A more detailed figure of the nail element in the numerical model would be useful;
4) Section 3.6 "Model comparison and validation" is valuable, but authors should underline the practical usefulness and benefit;
5) Conclusions highlight concepts already known in the field of slope stabilization using soil nails; authors are invited to highlight the new findings with respect to the available literature.
Some specific comments are reported in the attached annotated copy.

Author Response
REVIEWER 5:
Although the topic is interesting, nevertheless the paper does not present a high level of innovation. The content is more about applying a well-known stabilizing method to a case study slope, including a parametric numerical analysis, than actual research.
|
Reviewer’s comments |
Authors’ Response |
|
1) Table 1 needs amendments; |
Authors thank the reviewer for raising this important point. Values of d50 and d10 are corrected in this revised manuscript. |
|
2) It is not clear which is the model assumed at the interface between the nail and the surrounding soil; |
Thank you for the important comment. For ensuring proper soil-nail interaction a virtual interface factor of 0.1 is used which was multiplied by element thickness during mesh generation. Moreover, Rinter parameter was incorporated in the model, value of which was taken based on the findings of the previous studies. Please refer to line 208-212 and 216-220 for detailed explanation. |
|
3) A more detailed figure of the nail element in the numerical model would be useful; |
Please refer to Figure 2(b) for the detailed numerical model used for FE analysis. |
|
4) Section 3.6 "Model comparison and validation" is valuable, but authors should underline the practical usefulness and benefit; |
Authors are thankful to the reviewer for the comment. Authors agree with the reviewer that model comparison and validation part is valuable for any numerical study. Authors have attempted to compare the findings of the existing study with the relevant models who used different FEM and LEM methods for nailed slope to determine the accuracy of the model. And based on the analysis, it is observed that the results are in agreement with the findings reported in different scientific studies. As such the findings of the study, that is proposed optimum soil nail layout could be implemented in places where similar slope and soil exist to protect the hill slopes and stabilize it. |
|
5) Conclusions highlight concepts already known in the field of slope stabilization using soil nails; authors are invited to highlight the new findings with respect to the available literature. |
Conclusions have been updated to reflect the key findings. |
|
6) Table 4: Is it correct? 24cm only? |
The value is taken from the study of Rawat et al. (2013) where they developed a physical model and conducted the tests for soil nail stabilized slopes. |

Round 2
Reviewer 1 Report
The authors have successfully revised the manuscript. However, the authors should re - look into the following technicalities:
1. Line 115 - 119: Please provide an actual labelled image of the existing hill slopes in Rangamati. Please give a description about the slide height, spread of the fall and also the run - off length. This will validate the FE modelling of the slope and also help relate the accuracy of the modelled FE model.
2. Line 153 - 154: How did the authors calculated the backfill slope angle and slope angle of the actual slope. Describe briefly. The field study is too brief. Please elaborate as how the necessary parameters like slope angle, backfill angle, slope crest width, and overall slide height was recorded?
3. What about the facing of the nailed slope? The authors should mention about the facing scenario if this remediation is executed at the Rangamati slopes. Will it effect the nail inclination?
Author Response
Response to Reviewer’s Comments (Round-2)
Manuscript Title: Parametric Assessment of Soil Nailing on the Stability of Slopes Using Numerical Approach
Manuscript ID: geotechnics-1716234
The authors express their gratitude to all the reviewers for their valuable comments and suggestions to enhance the quality of the manuscript. Authors have tried their best to address and incorporate all the points recommended by the reviewers and authors believe that reviewers will appreciate the efforts of the authors and this present revised manuscript will be considered for publication in this reputed journal. The manuscript has been updated with “Track Changes” and also the new revisions (Round 2) are marked with blue marking.
REVIEWER 1:
|
Reviewer’s comments |
Authors’ Response |
|
1. Line 115 - 119: Please provide an actual labelled image of the existing hill slopes in Rangamati. Please give a description about the slide height, spread of the fall and also the run - off length. This will validate the FE modelling of the slope and also help relate the accuracy of the modelled FE model. |
Figure 1(a) has been added showing an actual labelled image of the existing hill slopes in Rangamati. A landslide failure is shown in Figure 1(b) which occurred in June 2017. From the failed slope, it can be observed that the failure starts from the crest of the slope, and then the soil mass moved downward and spread around the toe of the slope. Typically, the slide height was observed around 10 to 25 m whereas the spread of the fall was found in between 5m to 15 m. The numerical modeling has been validated based on the previous field studies. However, the authors couldn’t perform further validation against the failed slope since the numerical modeling was conducted for the nailed slope whereas the failure shown in the Figure 1(b) is a natural unreinforced slopes without soil nailing. |
|
2. Line 153 - 154: How did the authors calculated the backfill slope angle and slope angle of the actual slope. Describe briefly. The field study is too brief. Please elaborate as how the necessary parameters like slope angle, backfill angle, slope crest width, and overall slide height was recorded? |
The authors used Google Earth Pro software to determine the slope angles and other slope dimensions. The obtained data from Google Earth was also verified during the field study through field surveying. |
|
3. What about the facing of the nailed slope? The authors should mention about the facing scenario if this remediation is executed at the Rangamati slopes. Will it effect the nail inclination? |
Soil nails should be connected to a facing system at the excavation face or slope surface. The combinations of soil nails and facing should be dimensioned to sustain the expected maximum destabilizing force. There are different facing types are available for a nailed slope system. Shotcrete facing is typically less costly than the structural facing required for other wall systems [64]. Bioengineering technique using vegetation can be used as a substitute of structural facing which is effective for controlling erosion and sloughing of the soil at the slope surface [10, 65-67]. However, the use of this tech-nique should be limited to non-critical structures where large vertical and horizontal deformation are acceptable. For the bioengineering technique, facing of the nailed slope should not have any effect for the nail inclination. |
Reviewer 3 Report
The authors have taken into account all the comments by the reviewer.
Author Response
Response to Reviewer’s Comments (Round-2)
Manuscript Title: Parametric Assessment of Soil Nailing on the Stability of Slopes Using Numerical Approach
Manuscript ID: geotechnics-1716234
Authors express their gratitude to all the reviewers for their valuable comments and suggestions to enhance the quality of the manuscript. Authors have tried their best to address and incorporate all the points recommended by the reviewers and authors believe that reviewers will appreciate the efforts of the authors and this present revised manuscript will be considered for publication in this reputed journal. The manuscript has been updated with “Track Changes” and also the new revisions (Round 2) are marked with blue marking.
REVIEWER 3:
|
Reviewer’s comments |
Authors’ Response |
|
The authors have taken into account all the comments by the reviewer. |
The authors appreciate the comment of the reviewer. |
Reviewer 4 Report
Authors have incorporated all the suggested changes.
Manuscript can be accepted in the present form.
Author Response
Response to Reviewer’s Comments (Round-2)
Manuscript Title: Parametric Assessment of Soil Nailing on the Stability of Slopes Using Numerical Approach
Manuscript ID: geotechnics-1716234
Authors express their gratitude to all the reviewers for their valuable comments and suggestions to enhance the quality of the manuscript. Authors have tried their best to address and incorporate all the points recommended by the reviewers and authors believe that reviewers will appreciate the efforts of the authors and this present revised manuscript will be considered for publication in this reputed journal. The manuscript has been updated with “Track Changes” and also the new revisions (Round 2) are marked with blue marking.
REVIEWER 4:
|
Reviewer’s comments |
Authors’ Response |
|
Authors have incorporated all the suggested changes. Manuscript can be accepted in the present form. |
The authors appreciate the comment of the reviewer. |